# Openness Toward Organizational Change Scale (OTOCS): Validity evidence from Brazil and Portugal

Jorge Sinval[1,2,3,4], Vernon Miller[5], João Marôco[1] *

1 William James Center for Research, ISPA - Instituto Universitário, Lisbon, Portugal, 2 Business Research Unit (BRU-IUL), Instituto Universitário de Lisboa (ISCTE-IUL), Lisbon, Portugal, 3 Faculty of Philosophy, Sciences and Languages of Ribeirão Preto, University of São Paulo, São Paulo, Brazil, 4 Faculdade de Medicina, Universidade de Lisboa, Lisbon, Portugal, 5 Department of Communication and Department of Management, Michigan State University, East Lansing, Michigan, United States of America

* jpmaroco@ispa.pt

**Data Availability Statement:** All relevant data are within the manuscript and its Supporting information files.

**Funding:** This work was produced with the support of INCD and it was funded by FCT I.P. under the

## Abstract

Openness toward organizational change is central to employees' responses to organizations' strategic actions. This study aims to assess the validity evidence of the Openness Toward Organizational Change Scale (OTOCS) by examining the internal structure of the measure (e.g., dimensionality, reliability, measurement invariance) and its relations with other variables such as quality of work life, burnout, job satisfaction, and work engagement. A cross-sectional study was conducted using total sample of 1,175 workers, with 565 workers from Portugal and 610 from Brazil. The data provided satisfactory validity evidence based on the internal structure: the expected dimensionality was confirmed, acceptable levels of reliability were found, and measurement invariance was achieved among countries and sex. The measure also demonstrated satisfactory validity evidence based on the relations to other variables, being negatively associated with burnout and positively associated with work engagement, job satisfaction and quality of work. The OTOCS proved to be a relatively short self-report measure with satisfactory validity evidence to be used among Brazilian and Portuguese workers.

## Introduction

Certainty characterized the beginning of the 20th century, which deeply contrasts with the uncertainty at the end of the 20th and beginning of the 21st century [1]. Much uncertainty is associated with changes in institutions, including work organizations [2, 3]. It could be said that the only certainty in organizations is uncertainty, and effective managing of the unexpected consists in converting the uncertainty into a more orderly set of action [4]. As such, organizational change is a central feature of organizational behavior, and in turn in the lives of the individuals that compose them.

Organizations' change methods are often deemed ineffective and can cause resistance and burnout among workers [5]. Changes in work contexts, negative perceptions of key members,

project Advanced Computing Project CPCA/A0/ 7417/2020, platform Stratus. This research was supported by the Portuguese Foundation for Science and Technology (UID/PSI/04810/2019). The funders had no role in study design, data collection and analysis, decision to publish, or preparation of the manuscript.

**Competing interests:** The authors have declared that no competing interests exist.

the perceived lack of reciprocation of trust and respect, and negative perceptions of leadership/management contribute to the decline in organizational commitment [6]. As individuals' attitudes towards change play a fundamental role in the change process [7], conceptual work on measuring attitudes toward change appropriately continues to compel interest [8, 9]. One such employee attitude, openness to change, can be defined as positive affect and support for change and its consequences [10]. Openness is grounded in social exchange, where trust in both parties is given where deserved [11].

Openness toward change is of importance at various levels of organizational hierarchy. Research suggests that both top management openness and its sometimes corollary, trust in top management, can accentuate or moderate employee attitudes toward their work and their organization [12, 13]. Fulmer and Ostroff [14] suggest that trust in employees' immediate supervisors is associated with employee trust in top leadership, such that trust trickles upward but can also subsequently influences employee performance. By way of trust in top management, employee involvement and top management communication are indirectly related to organizational commitment [12]. Workers react positively if the organization actively includes them in the process of defining goals and maintains updates regarding the advancements towards those goals.

Top management openness can also be a major influence on organizational climate [15] as openness has its basis in affect and trust. Barsade and Gibson [16] note that affect permeates organizations as emotions (i.e., affective processes) create and support work motivation and that emotions spring to the fore when changes impact employees' work. Stouten et al.'s [17] extensive review of successful organizational change highlights the importance of open communication in the form of disclosing a change vision or explaining the vision. Emphasizing the change vision may enhance change efforts [18]. For example, the change vision might be communicated by a guiding coalition, which consists of a group of individuals within the organization that will shape, transmit, and sustain the change process [19]. The manner by which a proposed organizational change is communicated plays a fundamental role. Openness, honesty, and support at supervisory levels lead to greater employee receptivity [20, 21]. Judson [22] and Kanter et al. [20] note that it is not necessary to give complete details about the change. Yet, employees receiving a global perspective of the intended plan is essential. An understanding of the rationale behind the change process and its inherent tradeoffs is necessary to generate the desired support for the change effort [23].

Measures of employees' attitudes about organizational change are usually divided into four key constructs [24]: readiness for change; commitment to change; cynicism about organizational change; and openness to change. The readiness for change construct refers to the individual and organizational ability to carry out a successful change, the need for change, and the gains that an organization and its members can obtain from it [25]. Commitment to change is described as a force that drives the individual's actions toward the successful implementation of the organizational change plan [26]. Cynicism about organizational change pertains to pessimism towards implementing the organizational change plan due to the perception that those who responsible for its implementation are unmotivated, incompetent, or both [27]. The openness towards organizational change differs from the above constructs as it embraces the willingness to accommodate and accept change [10, 28].

## Openness toward change

Openness towards organizational change can be decisive to the successful implementation of new policies, processes, and structures in the workplace. Wanberg and Banas [28] found that employee optimism, perceived control, information received about changes and self-efficacy

for coping with changes were related to higher levels of change acceptance. Openness towards organizational change also depends on both individual variables (e.g., self-esteem, optimism, perceived control) and context-specific variables (e.g., information, participation, change self-efficacy, social support, personal impact). Furthermore, Lenberg et al. [7] reported that workers' feelings of participation in the change process, the knowledge about the intended changes outcomes, and their understating of the need for organizational change together impact on their attitudes towards said change. Occupational stressors, particularly poor work relationships, were negatively related to the attitudes towards change. In addition, highly stressed workers are prone to lower commitment and greater reluctance to accept change [29]. Openness to change also decreases when professionals' previous experiences with organizational change were negative and trust in management was low [30].

Of importance to organizational leadership, employees' attitudes toward change have important, long-lasting work-related consequences. Negative attitudes toward change impact employee job satisfaction, commitment to the organization, performance, and even their well-being [9, 28]. Changes in employees' work contexts, especially when they perceive changes to be unfair or negatively shaping their relationships or productivity, degrade their commitment to their employer [6]. Although employees' openness to organizational change has been at the forefront of concern for many organizational leaders [12, 17, 26], others are less attuned. As Oreg et al. [9] note, "managers are often oblivious to how change recipients will respond to the change and do not give enough thought to change recipients' perspectives." Consequently, top management's awareness of employees' openness toward change and its measurement is vital for organizational change efforts.

Although several survey instruments evaluating employee attitudes toward organizational change exist [9], this study focuses on the Openness Toward Organizational Change Scale [OTOCS; 10]. The original unidimensional scale was composed of five items, which was later modified by Wanberg and Banas [28] into a 7-item measure with two factors. For the propose of this study, the original measure was chosen since it gives a general measure of how open the workers are to the implementation of possible organizational changes in their work context. The 5-item version focuses on respondents' favorableness toward change in a relatively succinct manner. Its length is convenient when the practitioner/researcher wants to measure openness toward organizational change along with other constructs. The 5-item unidimensional measure offers a good balance between length and psychometric properties. The OTOCS assesses respondents' favorable attitudes toward change in the organization and the absence of resistance or rejection of change. Researchers have used the scale to measure specific changes [10] or potential changes [31].

The OTOCS has previously shown good validity evidence in different studies [28, 31–33]. In terms of its dimensionality, the measure has shown a stable number of items and dimensions. The original study suggested retaining five items (unidimensional structure) from a set of eight items [10]. Other authors used the pool of eight items from the original study and retained seven in a unidimensional structure [33]. Wanberg and Banas [28] created a 7-item modified version of the original OTOCS, suggesting two latent factors. Chawla and Kelloway [32] used the 8-item version assuming a single latent factor, and another study recently used the original 5-item version (as a one-factor structure) [31]. In terms of reliability, the OTOCS presented satisfactory evidence. Since its original study ($\alpha$ = .80) [10], the unidimensional version of the OTOCS presented good reliability estimates (in terms of internal consistency) that range from $\alpha$ = .83 [32] to $\alpha$ = .97 [33]. Since none of the existing studies assessed measurement invariance across groups (e.g., sex, country) within OTOCS, there is yet to be evidence of this psychometric property. The evidence pertaining to the relations to other variables is consistent, namely in terms of convergent evidence. Positive moderate correlations with work

engagement [31], organizational identification [10], trust, procedural justice, communication, and participation [32] were found. In addition, positive strong correlations with need for achievement and quality information [10] have also been reported.

This study focuses on the openness toward organizational change construct and its importance for the organizational literature. It adapts the OTOCS to Portuguese (in a single version both for Brazil and Portugal) and–if measurement invariance holds for both countries and sex—compares the scores between them. This study contributes to the literature by presenting the Portuguese adaptation of the OTOCS in keeping with the *Standards for Educational and Psychological Testing's* [34] recommendations for assessing instruments' validity evidence. The study seeks to verify the OTOCS' validity evidence based on the relation to other variables. It also presents a comparison between two countries that share many cultural similarities and have constant migration fluxes between them.

Based on the presented theoretical framework, the OTOCS measure is discussed, and the hypotheses related with its expected validity evidence, and comparisons are stated. First, the validity based on the internal structure is assessed (i.e., dimensionality, reliability, measurement invariance). Second, the validity evidence based on the relation to other variables is evaluated. Finally, the comparisons between sex and countries are established.

## Research hypotheses

Two different sources of validity evidence will be used to assess the OTOCS. The first source examines the internal structure of the instrument, which concerns the scale's dimensionality, reliability, and measurement invariance. Dimensionality refers to a cluster of items that correspond to an underlying latent variable [35]. That is, across multiple samples or contexts, the survey items evidence a consistent empirical relationship to a theoretical model. In the case of the OTOCS, the original scale suggests a unidimensional factor structure [10]. This unidimensionality was corroborated by a study with employees from a large international hospitality company's corporate office following organizational downsizing [33]. More recently, a study with Korean workers also confirmed the single-factor structure of the OTOCS [31]. Another modified scale with seven instead of five items proposed a two-factor solution, named "change acceptance" and "positive view of the changes" [28]. However, in the present study the original OTOCS was adapted, and it is assumed that its unidimensional structure (one latent factor which manifests itself in five indicators) will hold to the data (H1).

A second demonstration of validity pertains to the reliability of the instrument, or its lack of measurement error [35]. Reliability evidence can be obtained through different ways, one being its measure of internal consistency. An example of an internal consistency estimator is the Cronbach's α, which largely reflects the average inter-item correlation within the measurement items, tempered by the number of scale items [36]. The McDonald's ω [37] and the composite reliability (CR) [38] are other examples of internal consistency estimators. The original version [10] has exhibited evidence of reliability in terms of internal consistency (α = .80). Nunnally [39] suggests that for preliminary research with a psychometric instrument a threshold of .70 should be adequate. It is hypothesized that the OTOCS will present adequate internal consistency values (i.e., ≥.70) (H2).

A third evidence for validity is in measurement invariance, or where a instrument measures the unobserved construct in the same manner in each of the comparison groups [40]. To date, the authors are not aware of investigations that test for measurement invariance for sex or nationality in the OTOCS. Differences in these fundamental respondent characteristics are important as they address core elements in a sample from which to generalize. Furthermore, prior studies suggest that measurement invariance exists between samples of Portugal and

Brazil respondents [41, 42] and between sex in these countries [43, 44]. Since Portugal and Brazil are countries with cultural similarities, it is more likely to find measurement invariance between samples from these populations [45]. Consequently, it is expected that the OTOCS will be characterized by measurement invariance among countries and sex (H3).

Another source of validity focuses on evidence of a scale's relations to other variables [34]. Relevant constructs include affective dimensions such as quality of working life, burnout, job satisfaction, and work engagement. When the test scores relate closely to the same or similar constructs, convergent evidence is provided. On the other hand, discriminant evidence is provided when the instrument scores relate less closely to other constructs that should purportedly be less associated. Thus, measures demonstrate convergent evidence when they are positively correlated with constructs with which theory and prior research support positive relationships, but negatively correlated where theory and research portend that they should be negatively related [34]. In terms of positive associations, prior research indicates that higher levels of work engagement among employees lead to more successful interventions in the workplace [46]. Workers who are more work-engaged are more likely to develop job resources and consequently face organizational changes with greater success [47, 48]. Similarly, employees who perceive their working life is of high quality may be more open to change, a possible spillover effect of their job satisfaction with their work context and social exchange processes [49]. In fact, job satisfaction has been found to be positively related to openness to organizational change [28, 50].

In contrast, the openness toward organizational change is likely to be negatively associated with work burnout. Job burnout erodes employees' openness to organizational change in two ways [51]. Exhaustion, one component of burnout, stokes feelings of emptiness, and the need to rest while disengagement, a second component, is associated with distancing oneself from work and reinforcing negative attitudes [52]. Simply put, exhausted and disengaged workers are expected to be less open to potential changes in their work setting. Altogether, employee openness toward organizational change is expected to be positively associated with work engagement, quality of working life, and job satisfaction, but negatively correlated with burnout (H4).

Previous research suggests potential differences among sex or countries for constructs related to openness toward organizational change. Brazil and Portugal present different levels of uncertainty avoidance [53], which might influence their general responses to the OTOCS. With regard to potential differences in responses due to sex, openness to change was not significantly related to gender in two studies with workers from various occupations [30]. In line with these findings, sex was not significantly related to cognitive readiness for organizational change [54] or general readiness for change [55, 56]. However, as Morris and Venkatesh [57] and Venkatesh et al. [58] suggest, workplace innovations can differentially advantage male and female employees, influencing their willingness to adopt workplace innovations. Thus, it is hypothesized that different mean levels of openness toward organizational change should be observed due to participants' sex (H5.1) and between Brazilian and Portuguese workers (H5.2). It should be stated that potential statistically significant differences in latent mean scores among countries or sex do not abrogate the measurement invariance hypothesis. Latent mean scores should only be compared among different groups, after successfully obtaining measurement invariance [59].

## Method

### Sampling and data collection

The sample for this study involved 1,175 workers (S1 and S2 Datasets). This sample was composed of two samples, one involving multi-occupational workers in Brazil ($n$ = 610) and the

other involving multi-occupational workers in Portugal ($n$ = 565). The average age of the first sample was 35.11 ($SD$ = 10.13) with an average of 3.05 ($SD$ = 1.53) persons per household. Participants averaged working 9.73 ($SD$ = 8.61) years in their current job sector and 5.84 ($SD$ = 6.80) in their current organization, with 4.97 ($SD$ = 6.29) years and 1.85 ($SD$ = 2.06) promotions in their current job position. They also averaged working 3.38 ($SD$ = 3.89) years in previous jobs. Female participants constituted 67.23% of this sample.

For the second sample, participants averaged 35.83 ($SD$ = 9.76) years of age with 2.82 ($SD$ = 1.21) persons per household. They averaged working 11.23 ($SD$ = 9.69) years in their current job sector and 8.11 ($SD$ = 8.92) years in their current organization. They also worked an average of 6.14 ($SD$ = 7.05) years in their current position with 1.34 ($SD$ = 1.90) job promotions. These participants average working 2.34 ($SD$ = 3.03) years in past jobs. Female participants constituted 62.84% of this sample.

According to the *International Standard Classification of Occupations ISCO-08* [60], the samples from Brazil and Portugal mainly involved the "professionals" occupational group (see Table 1). ISCO-08 mentions that this group consist of teaching, science, engineering, business, administration, information and communication technology, or health professionals [60]. With respect to education, 74.39% of Brazil's sample and 83.07% of the Portuguese sample were college graduates.

The samples were based on non-probabilistic convenience sampling within the cross-sectional survey. The inclusion criteria required participants to be employed and able to read with

**Table 1. Occupational group, academic level, career and demographics statistics across countries.**

| | Brazil ($n$ = 610) | Portugal ($n$ = 565) | Total ($N$ = 1,175) |
|---|---|---|---|
| **Occupational groups %** | | | |
| Elementary Occupations | 1.55 | 1.81 | 1.68 |
| Plant and Machine Operators and Assemblers | 0.78 | 0.60 | 0.68 |
| Craft and Related Trades Workers | 2.14 | 2.22 | 2.18 |
| Skilled Agricultural, Forestry and Fishery Workers | - | - | - |
| Services and Sales Workers | 6.21 | 6.05 | 6.13 |
| Clerical Support Workers | 27.38 | 9.48 | 18.60 |
| Technicians and Associate Professionals | 8.74 | 12.90 | 10.78 |
| Professionals | 36.12 | 53.63 | 44.71 |
| Managers | 15.53 | 8.87 | 12.27 |
| Armed Forces Occupations | 1.55 | 4.44 | 2.97 |
| **Academic level %** | | | |
| High school, vocational education or less | 12.52 | 12.26 | 12.40 |
| Unfinished graduation | 13.09 | 4.67 | 8.93 |
| Graduation | 34.16 | 29.57 | 31.89 |
| Post-graduation (not master neither PhD) | 25.62 | 9.34 | 17.58 |
| Master | 9.49 | 38.52 | 23.82 |
| PhD | 5.12 | 5.64 | 5.38 |
| **Career and demographics** | | | |
| Working years in the current job sector $M(SD)$ | 9.73 (8.61) | 11.23 (9.69) | 10.46 (9.19) |
| Working years in the current job $M(SD)$ | 4.97 (6.29) | 6.14 (7.05) | 5.55 (6.69) |
| Working years in the current organization $M(SD)$ | 5.84 (6.80) | 8.11 (8.92) | 6.95 (7.99) |
| Sex (females) % | 67.23 | 62.84 | 65.07 |
| Age $M(SD)$ | 35.11 (10.13) | 35.83 (9.76) | 35.47 (9.95) |

easy access to a smartphone, tablet or computer, in order to use the digital platform where the survey questionnaire was deployed.

## Constructs and measures

**Openness Toward Organizational Change Scale (OTOCS).** The OTOCS is a psychometric instrument proposed by Miller, Johnson, and Grau [10]. This is a self-report measure composed of five items (two of them are reversed) which should be answered on a five-point ordinal scale, ranging from 1 - "To a very little extent" to 5 - "To a very great extent" (Table 2). It is intended to measure individuals' willingness to support organizational change and positive affect toward change (openness toward organizational change).

The original study of the OTOCS reported an acceptable reliability ($\alpha = .80$; $CR = .80$) and a unidimensional structure, with evidence of convergent validity (in terms of internal structure) nearly acceptable ($AVE = .45$) [10]. Additionally, the original study offered validity evidence based on the measure's relationship with other variables, such as organizational identification, role ambiguity, and quality information. Another study [61], using a longer version of this scale (8 items; including 3 items that were dropped in the original study), indicated evidence of internal consistency with an alpha of $\alpha = .90$.

To develop the Portuguese version (see Table 2), permission to adapt the original version was requested to one of the original authors [10]. The adaptation process was guided by *The ITC Guidelines for Translating and Adapting Tests* [62], and the items were adapted to Portuguese using the Orthographic Agreement which was created to be used by all countries that have Portuguese as their official language. The OTOCS' items were analyzed with various Portuguese and Brazilian researchers to guarantee semantic, idiomatic, and cultural equivalence in Portugal and Brazil. The OTOCS' original items were developed to measure the openness toward a specific implementation of organizational change (i.e., work teams). The original version presented in Table 2 shows the items from which the Portuguese was produced. In the Portuguese (Brazil and Portugal) version, the items were adapted without referring to a specific organizational change, instead, the items were adapted in reference to generic organizational changes in order to allow the maximization of the range of potential applications of the instrument. Using a sample of 15 workers from each country, a pilot study was conducted. The Portuguese version of the OTOCS used exactly the same wording in each item for both samples.

**Short Index of Job Satisfaction (SIJS).** The SIJS is a self-reported psychometric instrument [63] with five items (among which two are reversed) and one single latent factor (i.e., job satisfaction). Items are answered using an ordinal scale from 1 –"Strongly disagree" to 5

**Table 2. OTOCS original and Portuguese versions.**

| Item | Original OTOCS adapted [10] | | | | | Portuguese (Brazil and Portugal) version of OTOCS | | | | |
|---|---|---|---|---|---|---|---|---|---|---|
| | To a Very Little Extent | 2 | 3 | 4 | To a Very Great Extent | Em Reduzida Medida | 2 | 3 | 4 | Em Larga Medida |
| | 1 | | | | 5 | 1 | | | | 5 |
| 1 | I would consider myself to be "open" to changes to my work role. | | | | | Considero que estou "aberto" a mudanças que alterem as minhas funções no trabalho. | | | | |
| 2[R] | Right now, I am somewhat resistant to changes in my work. (R) | | | | | Neste momento, estou um pouco resistente a mudanças no meu trabalho. (R) | | | | |
| 3 | I am looking forward to the implementation of changes in my work role. | | | | | Estou ansioso pela implementação de mudanças nas minhas tarefas. | | | | |
| 4[R] | I am quite reluctant to consider changing the way I now do my work. (R) | | | | | Estou bastante relutante em mudar a forma como trabalho. (R) | | | | |
| 5 | From my perspective, the implementation of changes in my work will be for the better. | | | | | Na minha opinião, a implementação de mudanças no trabalho trará melhorias. | | | | |

*Note*: [R]–Reversed items.

–"Strongly agree." Previous studies revealed appropriate validity evidence of the SIJS, namely in terms of measurement invariance among countries (Portugal and Brazil) and sex, reliability of the scores, and dimensionality [44]. Examples of items are: "I feel fairly satisfied with my present job" and "Each day at work seems like it will never end" (reversed).

**The Quality of Working Life Scale (QWLS).** The psychometric instrument selected to measure the quality of work life was the QWLS [49]. It contains 16 items, which should be answered using an ordinal seven-point scale (from "1—Very Untrue" to "7—Very True"). The quality of work life is defined as the satisfaction of various needs at work [49]. The QWLS proposes quality of work life as a second-order latent factor, which in turn manifests itself through seven first-order factors: aesthetic needs (i.e., general aesthetics, creativity at personal and work level); knowledge needs (i.e., learning to improve skills at the job and professional level); actualization needs (i.e., fulfillment of the worker's potential as a professional and within the organization); esteem needs (i.e., appreciation and recognition of work outside and within the organization); social needs (i.e., leisure time off work and collegiality at work); economic and family needs (i.e., job security, pay, and other family needs); and health and safety needs (i.e., safety at work, preventive measures of health care, job-related benefits). The Portuguese version (Brazil and Portugal) presented measurement invariance across genders and countries, with evidence of reliability in terms of internal consistency and validity evidence based on the relationship with other variables [42]. Examples items are: "There is a lot of creativity involved in my job" (aesthetics needs), "I feel that I'm always learning new things that help do my job better" (knowledge needs), "I feel that my job allows me to realize my full potential" (actualization needs), "I feel appreciated at work at this organization" (esteem needs), "I have good friends at work" (social needs), "I have good friends at work" (economic and family needs), and "I feel physically safe at work" (health and safety needs).

**The Oldenburg Burnout Inventory (OLBI).** The Portuguese (Brazil and Portugal) version of the Oldenburg Burnout Inventory (OLBI) was used to measure burnout [43]. Burnout is defined as a syndrome consisting of constant exhaustion and negative attitudes regarding one's job due to stressors from the occupational environment [64]. It is conceptualized as a hierarchical latent variable, composed by two dimensions (first-order factors: exhaustion and disengagement). Exhaustion is defined as having feelings of emptiness and overtaxing regarding one's work and the need to rest, feelings of physical fatigue [65]. The disengagement latent factor is defined as one's developing negative and cynical behaviors and attitudes regarding one's job and creating distance from work in terms of content and object [66].

OLBI's Portuguese version includes 15 items and as uses five-point Likert scale (1 = "Strongly disagree," 2 = "Disagree," 3 = "Neutral," 4 = "Agree," 5 = "Strongly agree"). The exhaustion dimension comprises eight items, while the seven items were indicators of the disengagement factor. OLBI's Portuguese version [43] indicates measurement invariance between countries and sex, together with evidence of reliability in terms of internal consistency. It also suggested appropriate validity evidence based on relation to other variables, such as work engagement. Examples items are: "I always find new and interesting aspects in my work" (disengagement; reversed) and "There are days when I feel tired before I arrive at work" (exhaustion).

**Utrecht Work Engagement Scale (UWES-9).** The Utrecht Work Engagement Scale (short version) in Portuguese (i.e., Brazil and Portugal) [UWES-9; 41] was selected to measure work engagement through a hierarchical latent factor. Work engagement is described as energetic involvement with work [67] reflected in three first-order factors: dedication, absorption, and vigor [68]. Dedication is seen as showing feelings of significance, pride. enthusiasm and challenge, and a deep involvement with work [69]. Absorption is defined as being fully concentrated at work and happily engrossed, whereby the disconnection from work feels difficult

and the worker has the idea that time flies [70]. High energy in the workplace and mental resilience reflect vigor [70]. UWES-9 has nine manifest variables, with three items per each factor. An ordinal seven-point frequency scale was used to capture answers (0 = "Never," 1 = "A few times a year or less," 2 = "Once a month or less," 3 = "A few times a month," 4 = "Once a week," 5 = "A few times a week," 6 = "Always"). Evidence of reliability, measurement invariance between countries (Portugal and Brazil), and a hierarchical structure evidence (i.e. dimensionality) were found in the Portuguese version [41]. Examples of items are: "I am enthusiastic about my job" (dedication), "I feel happy when I am working intensely" (absorption), and "At my work, I feel bursting with energy" (vigor).

## Procedure

Workers from both samples completed the QWLS, the SIJS, the UWES-9, the OTOCS, and the OLBI psychometric instruments, together with demographics and career questions (i.e., working years in the current job sector, working years in the current job, working years in the current organization, sex, age). *LimeSurvey* [71] was the online platform selected to collect the data. Websites at two major universities, one each in Brazil and Portugal, deployed the survey to the participants. Information about the study was provided to all participants at the beginning of the survey. Participants were ensured that they were participating in a research study and that their employing organizations would not have access to individual responses. Electronic informed consent was also obtained from participants at the beginning of the survey. This study was approved by Committee of Ethics in Research with Human Beings (CEP) of the Faculty of Philosophy, Sciences and Letters of Ribeirão Preto, University of São Paulo, through "platform Brazil" of CONEP (National Commission of Ethics in Research). The written consent was obtained with the approval number 33301214.2.0000.5407. This study was also approved by the Ethics Committee of the Faculty of Psychology and Education Sciences of the University of Porto. The consent form stated, "the Ethics Committee of the Faculty of Psychology and Education Sciences of the University of Porto, having analyzed the research project. . . considers that it respects all ethical principles and ethical standards of research and therefore gives a favorable opinion."

## Data analysis

To conduct the statistical analysis the program *R* [72] through the integrated development environment, *RStudio* [73] was used. The *skimr* package [74] was utilized to produce the descriptive statistics. The skewness (sk) using "sample" method (i.e., sample skewness of the distribution) and the kurtosis using "sample excess" method (i.e., sample kurtosis of the distribution with a value of 3 being subtracted) were calculated using the *PerformanceAnalytics* package [75]. The coefficient of variation (CV) was estimated with the *sjstats* package [76], and the standard error of the mean (SEM) was calculated with the *plotrix* package [77]. The mode was computed by the *DescTools* package [78]. To evaluate the multivariate normality the *psych* package [79] was used to calculate the Mardia's multivariate kurtosis [80].

To obtain evidence about the originally proposed dimensionality of the measurement models, the confirmatory factor analysis (CFA) was used. The following goodness-of-fit indices were used: NFI (Normed Fit Index), TLI (Tucker Lewis Index), CFI (Comparative Fit Index), RMSEA (Root Mean Square Error of Approximation), SRMR (Standardized Root Mean Square Residual), and $\chi^2$/df (ratio Chi-Square and Degrees of Freedom). Estimates above.95 are considered acceptable for NFI, TLI, and CFI, whereas estimates smaller than 5 are considered acceptable for $\chi^2$/df [81]. Values below.08 are expected for SRMR and RMSEA [82]. It is known that for models with few degrees of freedom, RMSEA too often falsely indicates a poor

fitting [83]. RMSEA performance also depends on sample size and model misspecification [84, 85]. The SRMR presents evidence of being generally accurate across all conditions [86, 87]. Nevertheless, the evaluation of the goodness-of-fit of the tested models was based on the evidence provided by all mentioned indicators in conjunction. The package *lavaan* [88] was used to run the CFA analysis using the Weighted Least Squares Means and Variances (WLSMV) estimator [89]. The WLSMV was chosen because it does not require multivariate normality as an assumption and also because all items of the used psychometric instruments have an ordinal response scale.

The Average Variance Extracted (AVE) was estimated to test the evidence for convergent validity, as mentioned in Marôco [59] and Fornell and Larcker [38]. Satisfactory convergent validity evidence in terms of the internal structure was assumed for AVE $\geq$.5 [90].

To assess the evidence of reliability of the first-order factors, estimates of internal consistency were used: the CR, the $\alpha_{ordinal}$ and $\omega_{ordinal}$ [91]. Values of $\omega_{ordinal} \geq$.7, CR $\geq$.7 or $\alpha_{ordinal} \geq$.7 are considered indicative of acceptable reliability evidence. While the second-order latent factors had particular estimates of internal consistency: $\omega_{partial\ L1}$ (the proportion of variance explained by second-order factor after partialling the uniqueness of the first-order factor), $\omega_{L1}$ (the proportion of the second-order factor explaining the total score), and the $\omega_{L2}$ (the variance of the first-order factors explained by the second-order factor). Both second-order and first-order internal consistency estimates were calculated using the *semTools* package [91].

Using the theta-parameterization for categorical items through the *semTools* package [91] measurement invariance was evaluated comparing a group of four different models [92]: (a) configural invariance, (b) factor loadings, (c) thresholds of measured variables, and, (d) residual variances of observed variables.

The structural models (i.e., full structural equation models) were tested using *lavaan* to test validity based on relationships with other constructs. In the latent score means comparison, the Cohen's *d* [93] was used as effect size measure. The *doBy* package [94] was used to compute the raw score percentiles.

## Results

### Validity evidence based on the internal structure

**Items' distributional properties.** Table 3 presents various summary measures, a histogram, kurtosis (Ku), and skewness (Sk) for each of OTOCS' five items. The psychometric

**Table 3. Sample 1 and 2 descriptive statistics.**

| Item | M | SD | Min | Mdn | Max | Histogram | SEM | CV | Mode | Sk | Ku |
|---|---|---|---|---|---|---|---|---|---|---|---|
| **Brazil** | | | | | | | | | | | |
| Item 1 | 3.98 | 1.05 | 1 | 4 | 5 | | 0.04 | 0.26 | 5 | -0.86 | 0.02 |
| Item 2 | 3.99 | 1.14 | 1 | 4 | 5 | | 0.05 | 0.28 | 5 | -0.87 | -0.28 |
| Item 3 | 3.41 | 1.28 | 1 | 4 | 5 | | 0.05 | 0.37 | 4 | -0.37 | -0.90 |
| Item 4 | 3.86 | 1.19 | 1 | 4 | 5 | | 0.05 | 0.31 | 5 | -0.85 | -0.15 |
| Item 5 | 3.75 | 1.17 | 1 | 4 | 5 | | 0.05 | 0.31 | 5 | -0.73 | -0.24 |
| **Portugal** | | | | | | | | | | | |
| Item 1 | 4.03 | 0.91 | 1 | 4 | 5 | | 0.04 | 0.22 | 4 | -0.90 | 0.70 |
| Item 2 | 3.98 | 1.05 | 1 | 4 | 5 | | 0.04 | 0.26 | 5 | -0.97 | 0.37 |
| Item 3 | 3.23 | 1.11 | 1 | 3 | 5 | | 0.05 | 0.34 | 3 | -0.18 | -0.56 |
| Item 4 | 3.70 | 1.12 | 1 | 4 | 5 | | 0.05 | 0.30 | 4 | -0.55 | -0.39 |
| Item 5 | 3.58 | 0.99 | 1 | 4 | 5 | | 0.04 | 0.28 | 4 | -0.50 | 0.11 |

sensitivity and distributional properties of the items were analyzed with this information. Absolute values of Ku smaller than 7 and Sk smaller than 3 were indicative of no severe violations of the univariate normality that would recommend against the use of structural equation modeling [59, 95]. There was no severe normality violation in all items in both samples. The obtained distributional properties are suggestive of psychometric sensitivity. No multivariate normality was found, since Mardia's multivariate kurtosis presented high values for the two samples: Brazil (11.05, $p < .001$) and Portugal (15.87, $p < .001$). The OTOCS' five items had all possible values of the response scale in both samples, and, additionally, there were no outliers removed.

**Dimensionality.** The original OTOCS structure provided a poor fit for data obtained with the merged samples ($\chi^2(5) = 319.298$, $p < .001$, $\chi^2/df = 63.860$; $n = 1,175$; $CFI = .918$; $NFI = .917$; $TLI = .837$; $SRMR = .105$; $RMSEA = .231$; $P$(rmsea) $\leq.05) < .001$, 90% CI [.210;.253]). The modification indices were investigated, and a correlation path between item's 2 and item's 4 residuals was added to the model ($r = .459$, $p < .001$). The modified model presented an acceptable fit to the data from the merged samples ($\chi^2(4) = 96.363$, $p < .001$, $\chi^2/df = 24.090$; $n = 1,175$; $CFI = .976$; $NFI = .975$; $TLI = .940$; $SRMR = .058$; $RMSEA = .140$; $P$(rmsea) $\leq.05) < .001$, 90% CI [.117;.165]). All OTOCS' factor loadings were statistically significant ($p < .001$). There were no items removal. The minimum item loading for the OTOCS was .34. Consequently the fit for the separated samples was also acceptable (Fig 1), (Sample 1: $\chi^2(4) = 35.349$, $p < .001$, $\chi^2/df = 8.837$, $n = 610$; $CFI = .985$; $NFI = .983$; $TLI = .962$; $SRMR = .051$; $RMSEA = .113$; $P$(rmsea) $\leq.05) = .001$, 90% CI [.081;.149]; Sample 2: $\chi^2(4) = 64.087$, $p < .001$, $\chi^2/df = 16.022$, $n = 565$; $CFI = .967$; $NFI = .965$; $TLI = .918$; $SRMR = .067$; $RMSEA = .163$; $P$(rmsea) $\leq.05) < .001$, 90% CI [.129;.200]). RMSEA is known to be overestimated for models with few degrees of freedom [83].

The AVE value for the OTOC factor was .40, which is indicative of nearly satisfactory convergent validity evidence for this factor, obtaining similar individual values for each country ($AVE_{Brazil} = .39$; $AVE_{Portugal} = .40$). Globally, these results indicate nearly acceptable convergent validity evidence in terms of the internal structure of OTOCS.

**Reliability of the scores.** *Internal consistency.* To investigate the evidence of the reliability of the OTOCS, various internal consistency estimates were used. For sample 1, $\alpha_{ordinal}$, and $CR$

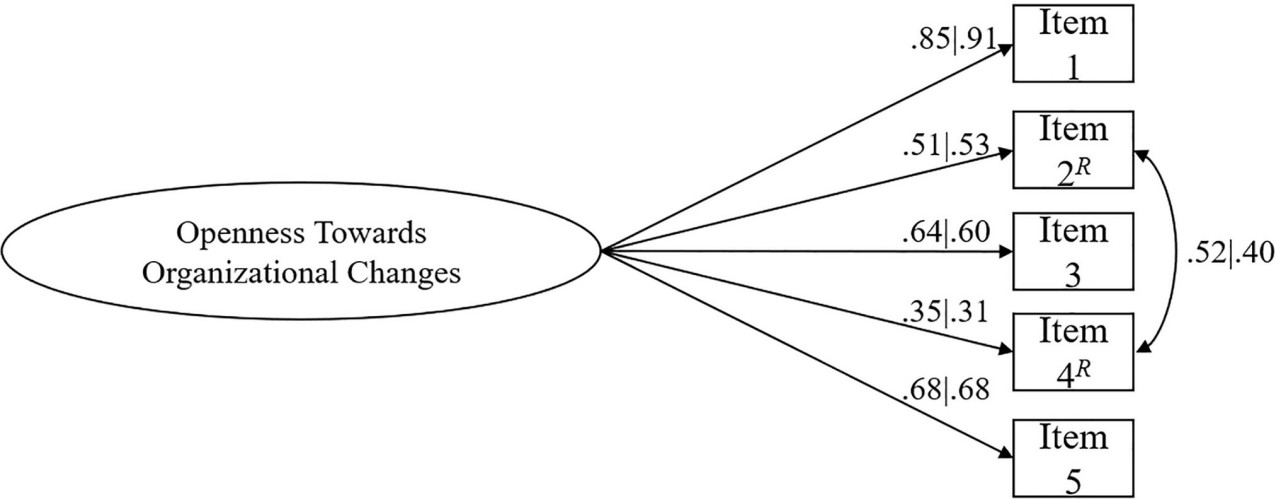

**Fig 1. OTOCS latent structure (five items).** Factor loadings and residual's correlations for each item are shown (Brazil | Portugal) and all of them were statistically significant ($p < .001$). $^R$Reversed items.

**Table 4. Internal consistency estimates for both samples.**

| Dimension | OTOCS | | | | | | | | |
|---|---|---|---|---|---|---|---|---|---|
| | Brazil | | | Portugal | | | Total | | |
| | α | ω | CR | α | ω | CR | α | ω | CR |
| Openness Toward Organizational Change | .76 | .66 | .75 | .75 | .66 | .75 | .75 | .66 | .75 |

presented internal consistency values above.7, although the $\omega_{ordinal}$ was below.7. For sample 2 $\alpha_{ordinal}$ and *CR* had values above.7, whereas $\omega_{ordinal}$ value was below.7. In global terms the obtained values were satisfactory for both samples (see Table 4). These results suggest an acceptable validity evidence in terms of the reliability.

**Measurement invariance.** Four nested models with indications of equivalence were developed to test if the same model holds in each sex and country. Multi-group confirmatory factor analyses (MGCFA) were used to assess the measurement invariance among sex and countries. Among the samples from Brazil and Portugal (see Table 5) metric invariance was supported by the $\Delta\chi^2$ criterion and scalar invariance was supported by the $\Delta$CFI $< .010$ criterion. Concerning measurement invariance among sex for each sample of the two countries, scalar invariance among sex in Brazil's sample was supported by both criteria. Full-uniqueness measurement invariance among sex in the Portuguese sample was supported by the $\Delta\chi^2$ criterion.

## Validity evidence based on relations to other variables

**Measurement models.** All the measurement models of each of the OTOCS criterion related variables were tested. The QWLS hierarchical model presented acceptable fit ($\chi^2(97) = 934.087$, $p < .001$, $\chi^2/df = 9.63$; $n = 1,095$; *CFI* = .990; *NFI* = .989; *TLI* = .988; *SRMR* = .065;

**Table 5. Measurement invariance between countries.**

| Model | $\chi^2$ | df | $\chi^2/df$ | CFI scaled | $\Delta\chi^2$ | $\Delta$CFI scaled |
|---|---|---|---|---|---|---|
| **Countries** | | | | | | |
| Configural | 99.436 | 8 | 12.430 | .934 | - | - |
| Metric | 102.665 | 12 | 8.555 | .943 | 3.373[ns] | .009 |
| Scalar | 133.578 | 26 | 5.138 | .936 | 38.373*** | .007 |
| Full uniqueness | 201.848 | 31 | 6.511 | .907 | 67.400*** | .029 |
| **Sex Brazil** | | | | | | |
| Configural | 35.058 | 8 | 4.382 | .968 | - | - |
| Metric | 35.593 | 12 | 2.966 | .975 | 1.293[ns] | .007 |
| Scalar | 55.416 | 26 | 2.131 | .967 | 22.594[ns] | .008 |
| Full uniqueness | 110.980 | 31 | 3.580 | .924 | 45.086*** | .043 |
| **Sex Portugal** | | | | | | |
| Configural | 68.089 | 8 | 8.511 | .904 | - | - |
| Metric | 72.837 | 12 | 6.070 | .919 | 4.921[ns] | -.015 |
| Scalar | 88.043 | 26 | 3.386 | .924 | 19.026[ns] | -.005 |
| Full uniqueness | 96.580 | 31 | 3.115 | .923 | 8.991[ns] | .001 |

*Note.*

[ns] $p > .05$;

* $p \geq .05$;

*** $p \geq .001$.

**Table 6. UWES-9, OLBI, QWLS and SIJS internal consistency estimates.**

| | UWES-9[2L] | OLBI[2L] | QWLS[2L] | | SIJS |
|---|---|---|---|---|---|
| ω | Vigor = .94 | Disengagement = .74 | Health and safety needs = .44 | | Job Satisfaction = .86 |
| | | | Economic and family needs = .64 | | |
| | Dedication = .91 | | Social needs = .46 | | |
| | | | Esteem needs = .78 | | |
| | | Exhaustion = .86 | Actualization needs = .90 | | |
| | Absorption = .88 | | Knowledge needs = .90 | | |
| | | | Aesthetics needs = .83 | | |
| $\omega_{partial\ L1}$ | .97 | .93 | .96 | | - |
| $\omega_{L1}$ | .94 | .86 | .91 | | - |
| $\omega_{L2}$ | .96 | .92 | .96 | | - |

*Note.*

[2L] –Second-order latent version.

$RMSEA = .089$; $P$(rmsea) $\leq.05$) $< .001$; 90% CI [.084;.094]). The UWES-9 second-order model obtained an acceptable fit, after constraining the variance of the first-order factor *dedication* to 0.01 in order to avoid negative variance ($\chi^2(25) = 441.365$, $p < .001$, $\chi^2/df = 17.65$; $n = 1,103$; $CFI = .998$; $NFI = .998$; $TLI = .997$; $SRMR = .041$; $RMSEA = .123$; $P$(rmsea) $\leq.05$) $< .001$; 90% CI [.113;.133]). The OLBI second-order model presented an acceptable fit to the data ($\chi^2(89) = 1,390.029$, $p < .001$, $\chi^2/df = 15.618$; $n = 1,104$; $CFI = .979$; $NFI = .977$; $TLI = .975$; $SRMR = .074$; $RMSEA = .115$; $P$(rmsea) $\leq.05$) $< .001$; 90% CI [.110;.121]). After adding one correlation among items' 3 and 5 residuals ($r = .408$, $p < .001$) the SIJS presented a very good fit to the data ($\chi^2(4) = 11.529$, $p = .021$, $\chi^2/df = 2.882$; $n = 1,102$; $CFI = .999$; $NFI = .999$; $TLI = .999$; $SRMR = .022$; $RMSEA = .041$; $P$(rmsea) $\leq.05$) $= .647$; 90% CI [.014;.070].

Globally, the reliability of the scores in terms of internal consistency of the first- and second-order latent factors for QWLS, UWES-9 and OLBI presented acceptable to very good evidence (Table 6). Although some of the QWLS first-order factors presented mediocre values, such result might be explained by the number of indicators in each first-order factor (only two or three items).

**Structural models.** The latent correlations were obtained through structural models. The fit of the structural model where work engagement was correlated with openness toward organizational change had a satisfactory fit to the data ($\chi^2(89) = 1,390.029$, $p < .001$, $\chi^2/df = 15.618$; $n = 1,103$; $CFI = .979$; $NFI = .977$; $TLI = .975$; $SRMR = .074$; $RMSEA = .115$; $P$(rmsea) $\leq.05$) $< .001$; 90% CI [.110;.121]) showing a large and positive correlation [96] between both measures ($r_{OTOC,\ WE} = .229$, $p < .001$). Concerning the structural model which correlated quality of work life with openness toward change, the fit to the data is acceptable ($\chi^2(180) = 1,653.144$, $p < .001$, $\chi^2/df = 9.184$; $n = 1,095$; $CFI = .984$; $NFI = .982$; $TLI = .981$; $SRMR = .071$; $RMSEA = .086$; $P$(rmsea) $\leq.05$) $< .001$; 90% CI [.083;.090]) presenting a medium and positive correlation [96] between both constructs ($r_{OTOC,\ QWL} = .138$, $p < .001$). The structural model that correlated job satisfaction with openness toward organizational change had an acceptable fit to the data ($\chi^2(32) = 354.970$, $p < .001$, $\chi^2/df = 11.093$; $n = 1,102$; $CFI = .983$; $NFI = .981$; $TLI = .976$; $SRMR = .073$; $RMSEA = .096$; $P$(rmsea) $\leq.05$) $< .001$; 90% CI [.087;.105]) the correlation was small [96] and positive ($r_{OTOC,\ JS} = .097$, $p < .001$). Finally, regarding the latent correlation between openness toward organizational change and burnout the structural model presented an acceptable fit to the data ($\chi^2(166) = 2,164.103$,

**Table 7. Quartiles, means, and standard deviations for countries and sex.**

| Dimension | Sex | | | | | | | | | | | | | | |
|---|---|---|---|---|---|---|---|---|---|---|---|---|---|---|---|
| | Brazil | | | | | | | | | | | | | | |
| | Female (*n* = 355) | | | | | Male (*n* = 173) | | | | | Total (*n* = 610)* | | | | |
| | *M* | *SD* | 25 | 50 | 75 | *M* | *SD* | 25 | 50 | 75 | *M* | *SD* | 25 | 50 | 75 |
| Openness toward organizational change | 3.84 | 0.76 | 3.20 | 3.80 | 4.40 | 3.81 | 0.75 | 3.20 | 3.80 | 4.40 | 3.80 | 0.77 | 3.20 | 3.80 | 4.40 |
| Dimension | Portugal | | | | | | | | | | | | | | |
| | Female (*n* = 323) | | | | | Male (*n* = 191) | | | | | Total (*n* = 565)* | | | | |
| | *M* | *SD* | 25 | 50 | 75 | *M* | *SD* | 25 | 50 | 75 | *M* | *SD* | 25 | 50 | 75 |
| Openness toward organizational change | 3.73 | 0.71 | 3.40 | 3.80 | 4.20 | 3.66 | 0.68 | 3.20 | 3.60 | 4.20 | 3.70 | 0.68 | 3.20 | 3.80 | 4.20 |

*Note.*

*—Some subjects did not inform their sex.

$p < .001$, $\chi^2/df$ = 13.037; $n$ = 1,104; *CFI* = .970; *NFI* = .968; *TLI* = .966; *SRMR* = .078; *RMSEA* = .104; $P$(rmsea) $\leq$.05) < .001; 90% CI [.101;.108]) the obtained correlation was large [96] and negative ($r_{OTOC, B}$ = -.232, $p < .001$). Altogether, the validity evidence based on relations to other variables was satisfactory, and the direction and magnitude of the correlations occurred as expected.

## Openness toward organizational change comparisons among sex and country

Comparisons of OTOCS' overall mean scores between the groups (countries and sex) were performed since the measurement models evidenced scalar invariance. The latent means were compared within the structural equation modeling framework. The comparison between workers from Portugal and Brazil revealed statistically significant differences ($\Delta\chi^2_{scaled}(1)$ = 4.639; $p$ = .031; $d$ = 0.274), with the sample from Portugal presenting lower levels of openness toward organizational change. Moreover, comparisons by participants' sex did not reveal statistically significant differences ($\Delta\chi^2_{scaled}(1)$ = 0.933; $p$ = .334; $d$ = 0.120). For comparative proposes for future OTOCS studies, the means and quartiles for sex and countries are presented in Table 7.

## Discussion

It is relatively easy for top management to assume planned organizational changes will enhance the workplace and organizational viability. Statements such as, ". . .organizations must create a healthy discomfort with the status quo" [97], fail to appreciate employee reservations when plans for reorganization, downsizing, or new operations are announced. Employees are no doubt mindful of unintended, less than stellar outcomes from prior organizational change efforts [5]. Established measures of employee attitudes toward change may be helpful for researchers and practitioners alike to understand organizational member sentiments and factors contributing to their receptivity or reservations about impending changes.

The obtained findings herein provide satisfactory validity evidence based on relationships with other variables and on the scale's internal structure [34]. Findings related to the first hypothesis (H1) suggest acceptable goodness-of-fit of the data to the OTOCS's original structure. Thus, H1 was supported, confirming the original structure of five indicators and one latent factor (i.e., openness toward organizational change). The maintenance of the original dimensionality is important since it provides evidence of the measure's transcultural stability.

Moreover, these analyses attest to the degree to which the relationships among the OTOCS items conform to the dimension that is being measured and on which the proposed test score interpretations are based [34]. The measurement of the latent variable is a prerequisite to the analysis of causal relations among latent variables [98]. In this case, the minimum λ value was lower than desirable (item 4). However, the retention of the item did not cause a poor fit to the model, though the loading of this particular item should be investigated in future analysis with independent samples from the same population. A modification was added (i.e., a correlation among item 2's and item 4's residuals), which improved the model's fit. Adding atheoretical paths based on modifications indices is inappropriate [99]. However, the included modification is justifiable as both items belong to the same factor (i.e., residuals might be associated) and both items were negatively worded [100]. The use of reversed items has trade-offs, and the practice should be used with caution [100, 101]. By correlating those items' residuals means, a portion of variance that the model is not capable of explaining is related to responses to both items. The lower λ value may stem from several factors, including a different interpretation of the items [102]. The obtained convergent validity in terms of the internal structure was nearly acceptable, with the mean of the loadings being .60.

The evidence provided by the reliability of the scores was satisfactory in terms of internal consistency estimates, particularly the α and *CR* values which were above .70. The ω values were slightly below that which was expected. All the estimates' values were homogeneous between countries. Altogether, evidence generally supports H2. The values of the ordinal α were near the ones observed in the original study of the measure (α = .80) [10] and in line with other studies with measures based on the original scale [30, 103]. Other studies using measures based on the original OTOCS were in line with the observed reliability scores [28, 104].

Findings here also support H3 as scalar invariance was observed among the Portuguese and Brazilian samples and among sex for each country. None of the previous studies using the original OTOCS or measures based on the OTOCS investigated the measurement invariance among countries or sex. These findings may be particularly useful since they allow comparisons of scores between Portuguese and Brazilian samples and between participant sex. Scalar measurement invariance also allows comparisons between mean scores. Measurement invariance among Portugal and Brazil has been found before with other psychometric instruments [41–43].

The relationship of the OTOCS to other variables offers convergent evidence [34]. Findings here revealed statistically significant associations with other variables occurred as expected, both in terms of magnitude and in terms of direction in support of H4. That is, the OTOCS is positively associated with quality of work life, work engagement, and job satisfaction, and negatively associated with job burnout. The positive views of organizational change are related with the perception of resource improvements which in turn can be associated with work engagement [105]. Relatedly, recent research suggested that long-term attitude-to-change can be predicted by work engagement [106]. Additionally, job satisfaction can be associated with openness toward organizational change, since lower levels of change acceptance are related to lower levels of job satisfaction [28]. The OTOCS scores' correlation with job satisfaction scores were slightly lower than the ones found among the same dimensions in other studies [28, 104]. As expected, perceptions of their quality of work life was positively correlated with OTOC scores. Employees may be more supportive of organizational changes if they have positive perceptions of what the organization gives them in terms of their need for job satisfaction and well-being [107–109]. Similarly, Franco et al. report [107] that acceptance of change positively predicted perceptions of well-being.

In addition, participants' responses to the OTOCS were negatively correlated with burnout in keeping with other studies examining employee perceptions of openness of communication

and openness to change and burnout [51]. For instance, Vakola and Nikolaou [29] found a negative correlation between attitudes towards organizational change and job stress perceptions ($r$ = -.20), which is in line with the magnitude and direction of the correlation found between OTOCS scores and OLBI scores in the current study. It is important to note that in this study, burnout levels were associated with the general possibility of organizational change, not before or after the implementation of change. Given that increased stress among employees is a common reaction of the organizational change [9], it is important for firms to assess employee attitudes toward potential, unspecified changes [110]. It almost goes without saying that the existence of high stress levels is never a good starting point for organizational change [18]. On top of possible resistance to change, strained workers have less resources to produce positive views and participate in planned organizational change [17].

Finally, statistically significant different latent means of openness toward organizational change were observed between countries, providing support for H5.1. These results might be explained by cultural differences since Portugal and Brazil have a cultural background where uncertainty avoidance clearly differs among them [111]. Portugal scores the second highest in uncertainty avoidance while Brazil is placed in 31$^{st}$-32$^{nd}$ positions interval in a list of 76 countries. In line with Hofstede et al.'s [111] work, results from this study show that Portugal scores were lower in terms of openness toward organizational change in comparison with the ones from Brazil. Such comparison is noteworthy since the structure of the OTOCS measure is invariant (at least at the scalar level) between both countries [112]. Finally, no statistically significant differences were found between the openness toward organizational change among sex. Thus, the data did not support H5.2. As noted earlier, gender was also not being related to openness toward organizational change in several earlier studies [30, 54–56].

## Limitations and future research

This study focuses on employees' openness to change in their organizations, irrespective of current or future planned changes in participants' organizations. It can be argued that nowadays some degree change in work processes and unit configurations occurs constantly, and all organizations must be prepared to deal with constant change [5]. However, this investigation did not gather data about past organizational history of prior planned change or current specific changes, which would be useful to develop insights about employees' positive or negative views about organizational change [113]. At the same time, there is considerable value in measuring employees' openness to change when a planned change is not ongoing or imminent [110]. Information about their attitude toward change in general can provide insights regarding employee overall adaptability, past management actions, variation in employee attitudes from one unit to another, or work unit flexibility.

The current study's research design also compared responses to the OTOCS and related constructs between two countries with different cultures, though connected through a similar language. Albeit valuable, the scope of this study was restricted. Consequently, future studies should explore the OTOCS' psychometric properties across a more diverse set of countries, cultures, and demographic groups. Investigations with a more complex nomological network of related constructs could provide broader evidence convergent and discriminant evidence.

Since this study presents a cross-sectional design, no temporal stability of the measure structure and scores is possible. As a starting point, future studies should approach longitudinally the OTOCS's structure (i.e., longitudinal measurement invariance) and its scores. Obtaining predictive criterion evidence is also encouraged, which will allow to assess OTOCS' scores against a criterion score (e.g., organizational change engagement behaviors, trust in top

management) [34]. Tests of the scale's predictive criterion evidence will be an important step in assessing its long-term value in organizational studies.

## Conclusions

Organizational change is omnipresent, and employee openness toward planned changes an important element to achieving organizational goals. The measurement of openness toward organizational change should be rigorous, as all analyses of constructs should be, or they will not be useful. The OTOCS is a psychometric instrument that appears as a good instrument to measure openness to change in organizational settings, presenting good to acceptable validity evidence from various sources. Being a short instrument (i.e., five items) facilitates its use in conjunction with other survey measures.

## Supporting information

**S1 Dataset. The dataset used.**
(CSV)

**S2 Dataset. The dataset codebook.**
(CSV)

## Acknowledgments

The authors would like to thank the Portuguese national occupational health program of the Directorate-General of Health (DGS) for the scientific sponsorship.

## Author Contributions

**Conceptualization:** Jorge Sinval, João Marôco.

**Data curation:** Jorge Sinval.

**Formal analysis:** Jorge Sinval.

**Funding acquisition:** João Marôco.

**Investigation:** Jorge Sinval.

**Methodology:** Jorge Sinval.

**Resources:** Jorge Sinval.

**Software:** Jorge Sinval.

**Supervision:** Vernon Miller, João Marôco.

**Validation:** Jorge Sinval, Vernon Miller.

**Visualization:** Jorge Sinval.

**Writing – original draft:** Jorge Sinval.

**Writing – review & editing:** Jorge Sinval, Vernon Miller, João Marôco.

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
