## [Decision Letter · Decision Letter 0]

14 Jan 2021

PONE-D-20-29649

Openness Toward Organizational Change Scale (OTOCS): Validity Evidence from Brazil and Portugal

PLOS ONE

Dear Dr. Marôco,

Thank you for submitting your manuscript to PLOS ONE. After careful consideration, we feel that it has merit but does not fully meet PLOS ONE’s publication criteria as it currently stands. Therefore, we invite you to submit a revised version of the manuscript that addresses the points raised during the review process.

In general, we found that the manuscript contributes to increase the knowledge about the construct of Openness Toward Organizational Change Scale. However, both reviewers raised several theoretical as well as methodological concerns.

The most critical issues concerned the theoretical background that justify the focus of the study and the method used and the conclusions.

Based on my own reading, as well as the input of the reviewers, I see enough promise/potential to move forward with your manuscript and invite a revision.

We look forward to receiving your revised manuscript.

Kind regards,

Mariagrazia Benassi

Academic Editor

PLOS ONE

Journal Requirements:

Reviewers' comments:

Reviewer's Responses to Questions

**Comments to the Author**

1. Is the manuscript technically sound, and do the data support the conclusions?

Reviewer #1: Yes

Reviewer #2: Partly

2. Has the statistical analysis been performed appropriately and rigorously? 

Reviewer #1: Yes

Reviewer #2: No

3. Have the authors made all data underlying the findings in their manuscript fully available?

Reviewer #1: Yes

Reviewer #2: No

4. Is the manuscript presented in an intelligible fashion and written in standard English?

Reviewer #1: Yes

Reviewer #2: No

5. Review Comments to the Author

Reviewer #1: Dear colleagues,

I’ve recently had the opportunity to review the manuscript “Openness Toward Organizational Change Scale (OTOCS): Validity Evidence from Brazil and Portugal”. I have appreciated your manuscript that I consider as consistent and well done.

I have some concerns and comments. I hope they are useful to you to move forward.

Please describe with more details:

1) the construct of Openness Toward Organizational Change Scale as measured by OTOCS, in relationship with the item contents.

2) Why do you adopt the 5-item version?

3) the psychometric qualities of OTOCS that literature presents

4) why do you accept the models even if the RMSEA indexes are very unfair?

5) why do you assume Weighted Least Squares Means and Variances (WLSMV) estimator?

6) some lines of organization interventions when there is a low level of Openness Toward Organizational Change

Reviewer #2: I am sympathetic to value of a Portuguese language version of commonly used scales in organizational research. The authors have provided evidence of the psychometric properties of a variety of commonly used scales including Engagement and Openness to Change. What puzzles me is this: why is the focus of this paper Openness to Change when none of the other indicators used pertain to that topic.

The present version uses a two-country sample of Portuguese speaking individuals and surveys them using a questionnaire that contains a mixed bag of common scales in organizational research. Yet the focus of the paper is the phenomenon of Openness to Change and the suitability of this particular translation of a common scale to assess it. The convenience samples in Brazil and Portugal do not appear to involved organizations undergoing a systematic or planned organizational change. Despite the assertions of the authors, Openness to Change is a construct largely targeting change recipients to understand the role of individual predispositions in shaping their change responses. Not only is there no evidence that change is occuring in the organizations from which participants are derived, but there is no measure of change-related perceptions, processes/interventions or outcomes.

This paper could easily have been written with a focus on engagement, for example, and then the psychometric properties of that scale could be the central story. If change-related experiences are not assessed we do not know whether the Openness to Change measure functions in the typical context in which it is used. It is true that engagement and stress are measured in lots of contexts and it may be better to refocus the present report on Portuguese translations of common organizational measures, and downplay the focus on change, since the sampling strategy is not tied to change experiences.

Methodologically, I am concerned that there is no change context studied, limiting the information provided regarding the explanatory power of the Portuguese version of Openness to Change for change research. I also am concerned with idiosyncratic adjustments made in the interdependencies among items in order to engineer good fit. I recognize that tools are available in various structural equation programs to enhance conventional indicators, but this is a practice not widely accepted. It would be better to report the non-engineered and engineered indicators of fit and talk through the sensitivity of observed effects to these adjustments.

So, I am suggesting that a connection between organizational change and the present survey and sample design is not yet established. Can you make a better case, perhaps through additional change-related measures, or further evidence regarding the change experience of your sample?

I suggest as an alternative a focus on the Portugeuse language assessment of common organizational survey measures. I do not know this literature but in trying to find contributions that can be made by the current data set that possibility comes to mind.

6. PLOS authors have the option to publish the peer review history of their article (what does this mean?). If published, this will include your full peer review and any attached files.

Reviewer #1: No

Reviewer #2: No

---

## [Author Response · Author response to Decision Letter 0]

16 Mar 2021

Manuscript ID: PONE-D-20-29649

“Openness Toward Organizational Change Scale (OTOCS): Validity Evidence from Brazil and Portugal”

Reviewer 1

###########################################################

Reviewer 1’s comments

###########################################################

I’ve recently had the opportunity to review the manuscript “Openness Toward Organizational Change Scale (OTOCS): Validity Evidence from Brazil and Portugal”. I have appreciated your manuscript that I consider as consistent and well done.

I have some concerns and comments. I hope they are useful to you to move forward. 

Authors’ response 

Thank you for your careful reading of the manuscript. Responses to your questions and concerns are below.

###########################################################

Reviewer 1’s comments

###########################################################

Please describe with more details:

1) the construct of Openness Toward Organizational Change Scale as measured by OTOCS, in relationship with the item contents. 

Authors’ response 

We added to pages 6-7:

“The original five-item scale assesses respondents’ favorable attitudes toward change in the organization and the absence of resistance or rejection of change. Researchers have used the scale to measure specific changes [10] or potential changes [31].”

Please describe with more details:

###########################################################

Reviewer 1’s comments

###########################################################

2) Why do you adopt the 5-item version? 

Authors’ response 

We added (page 6):

“The 5-item version focuses on respondents’ favorableness toward change in a relatively succinct manner. Its length is convenient when the practitioner/researcher wants to measure openness toward organizational change along with other constructs. The 5-item unidimensional measure offers a good balance between length and psychometric properties.”

###########################################################

Reviewer 1’s comments

###########################################################

Please describe with more details:

3) the psychometric qualities of OTOCS that literature presents 

Authors’ response 

Psychometric properties of the OTOCS obtained from other studies was added to the manuscript (page 7):

“The OTOCS has previously showed good validity evidence in different studies [28,31–33]. In terms of its dimensionality, it has shown a stable number of items and dimensions. The original study suggested retaining five items (unidimensional structure) from a set of eight items [10]. Other authors used the pool of eight items from the original study and retained seven of them (unidimensional structure) [33]. Wanberg and Banas [28] used a 7-item modified version of the original OTOCS, suggesting two latent factors. Chawla and Kelloway [32] used the 8-item version assuming a single latent factor. Recent research has implemented the original 5-item version (one-factor structure) [31]. In terms of reliability, the OTOCS presented good evidence. Since its original study (α = .80) [10], the unidimensional version of the OTOCS presented good reliability estimates (in terms of internal consistency), ranging from α = .83 [32] to α = .97 [33]. Since none of the existing studies assessed measurement invariance across groups (e.g., sex, countries) within OTOCS, there is no evidence of any kind regarding this type of psychometric property. The evidence pertaining to the relations to other variables is consistent, namely in terms of convergent evidence. Positive moderate correlations with work engagement [31], organizational identification [10], trust, procedural justice, communication, and participation [32] were found. And positive strong correlations with need for achievement and quality information [10] were also reported.”

###########################################################

Reviewer 1’s comments

###########################################################

Please describe with more details:

4) why do you accept the models even if the RMSEA indexes are very unfair? 

Authors’ response 

The RMSEA is one of the various possible goodness-of-fit indices with some known particularities, namely, it depends on sample size and model misspecification, and model degrees of freedom [1,2]. The SRMR presents evidence to be generally accurate across all conditions [3,4]. Nevertheless, other goodness-of-fit indices were used (CFI, TLI, NFI) in conjunction to assess models’ adequacy.

The OTOCS model presents a particularly small number of degrees of freedom and thus can be more exposed to the RMSEA under optimal performance in models with few degrees of freedom.

We added this, information to the data analysis section:

“It is known that for models with few degrees of freedom RMSEA too often falsely indicates a poor fitting [81]. RMSEA performance also depends on sample size and model misspecification [82,83]. The SRMR presents evidence of being generally accurate across all conditions [84,85]. Nevertheless, the evaluation of the goodness-of-fit of the tested models will be based on the evidence provided by all mentioned indicators in conjunction.”

###########################################################

Reviewer 1’s comments

###########################################################

Please describe with more details:

5) why do you assume Weighted Least Squares Means and Variances (WLSMV) estimator? 

Authors’ response 

We added on page 19:

“The WLSMV was chosen because it does not require multivariate normality as an assumption and also because all items of the used psychometric instruments have an ordinal response scale.”

###########################################################

Reviewer 1’s comments

###########################################################

Please describe with more details:

6) some lines of organization interventions when there is a low level of Openness Toward Organizational Change

Authors’ response 

The importance of assessing employees’ openness toward change at anytime, even if a planned change is not imminent, is fairly well expressed by Malik and Garg [5]:

“…technological transformations put immense pressure on employees since they encounter frequent changes in roles and responsibilities and reskill constantly to gear up with new lines of reporting and modifications in IT operations. This necessitates these firms to prepare their employees to adopt to change rather than retaliate against it. IT organizations thus need to ensure that their employees adapt effectively to these change processes and elicit affective commitment to change.”

Although Malik and Garg focus on the tech industry, knowledge of employees’ orientation toward planned change or incremental changes related to adapting to everyday work challenges can provide insight into a range of factors – past management actions, employees’ attitude formation, work unit flexibility, etc.

As so, we added the following lines to the manuscript:

Page 29-30

“It is important to note in this study burnout levels were associated with the general possibility of organizational change, not before or after the implementation of change. Given stress among employees is a common reaction of the organizational change [9], it is important for firms to assess employee attitudes toward such changes [101]. It almost goes without saying that the existence of high stress levels is never a good starting point for organizational change [18]. On top of possible resistance to change, strained workers have less resources to produce positive views and participate in planned organizational change [17].”

Page 31

“At the same time, there is considerable value in measuring employees’ openness to change when a planned change is not ongoing or imminent [101]. Information about their attitude toward change in general can provide insights regarding employee overall adaptability, past management actions, variation in employee attitudes from one unit to another, or work unit flexibility.”

#####################################

#####################################

#####################################

Reviewer 2

###########################################################

Reviewer 2’s comments

###########################################################

I am sympathetic to value of a Portuguese language version of commonly used scales in organizational research. The authors have provided evidence of the psychometric properties of a variety of commonly used scales including Engagement and Openness to Change. 

Authors’ response 

Thank you for your careful reading of the manuscript. Responses to your questions and concerns are below.

###########################################################

Reviewer 2’s comments

###########################################################

What puzzles me is this: why is the focus of this paper Openness to Change when none of the other indicators used pertain to that topic. The present version uses a two-country sample of Portuguese speaking individuals and surveys them using a questionnaire that contains a mixed bag of common scales in organizational research. Yet the focus of the paper is the phenomenon of Openness to Change and the suitability of this particular translation of a common scale to assess it. 

Authors’ response 

Your comment raises two important issues, one of which is brought up several times.

First, one issue concerns the paper’s orientation vis a vis impending/current change processes.

There is great value in scales that assess employees’ attitudes about change when planned change is ongoing or imminent. In these contexts, the openness to change scale may provide insight into employee responses during a change or provide a comparison of attitudes from one time point and a second time point. 

There is also considerable value in measuring employees’ openness to change when a planned change is not ongoing or imminent. Information about their attitude toward change in general can provide insights regarding employee overall adaptability, past management actions, variation in employee attitudes from one unit to another, work unit flexibility, etc.

A number of authors have measured openness to changes when organizational change was not imminent. Here are a few: 

Chai et al. [6] – use the Miller et al., [7] scale, Korean sample, no imminent change context;

Danish et al., [8] – use the Wanberg and Banas [9] scale that incorporates the Miller et al., [7] scale, Indian sample, no imminent change context;

Malik and Garg [5] – use the Herscovitch and Meyer [10] affective commitment to change scale, Indian sample, no imminent change context;

Gori and Topino [11] – use the Di Fabio and Gori [12] scale on acceptance of change, Italian sample, no imminent change context.

Second, the “mixed bag of common scales in organizational research” provides convergent evidence with those constructs. Importantly, the relationship of the OTOC measure with job satisfaction, quality of work life, burnout, and work engagement scales provide a point of comparison to prior research, in keeping with the Standards for Educational and Psychological Testing [13]. A natural next step for research is to assess concurrently the OTOCS’ convergent and discriminant validity, which we include in the Discussion section.

###########################################################

Reviewer 2’s comments

###########################################################

The convenience samples in Brazil and Portugal do not appear to involved organizations undergoing a systematic or planned organizational change. Despite the assertions of the authors, Openness to Change is a construct largely targeting change recipients to understand the role of individual predispositions in shaping their change responses. Not only is there no evidence that change is occuring in the organizations from which participants are derived, but there is no measure of change-related perceptions, processes/interventions or outcomes.

Authors’ response 

As noted above, we did not intend to test the effectiveness of interventions based on their collaborators’ levels of OTOC.

An important element in the paper is the responses of Portuguese and Brazilian workers to the OTOCS, given their cultural similarities and differences.

The reviewer brings up an interesting aspect, which was added to the future research agenda with the OTOCS. The use of criterion measures (i.e. test-criterion evidence) can be assessed with two approaches (predictive or concurrent). Predictive evidence (indicates the strength of the relationship between test scores and criterion scores that are obtained at a later time) e.g. OTOCS relationships with measures of the effectiveness of organizational change. Concurrent evidence (obtains test scores and criterion information at about the same time) e.g., relationships of OTOCS with change-related perceptions.

###########################################################

Reviewer 2’s comments

###########################################################

This paper could easily have been written with a focus on engagement, for example, and then the psychometric properties of that scale could be the central story. If change-related experiences are not assessed we do not know whether the Openness to Change measure functions in the typical context in which it is used.

Authors’ response 

We appreciate the reviewer’s focus on change-related experiences as well as the value of assessing the relationship of engagement along with the OTOCS. As noted earlier, we note the importance of convergent validity investigations as both a limitation of the current study and a direction for future research. 

A chief goal of the manuscript was to assess the psychometric properties of the instrument (i.e., measurement model).

Another aim of future investigations is to examine employee openness to change in Portuguese-language countries in a longitudinal research design under relevant interventions. For instance, longitudinal measurement invariance can be tested to assess the stability of the measurement structure of the OTOC, a point now included as future research.

###########################################################

Reviewer 2’s comments

###########################################################

It is true that engagement and stress are measured in lots of contexts and it may be better to refocus the present report on Portuguese translations of common organizational measures, and downplay the focus on change, since the sampling strategy is not tied to change experiences.

Methodologically, I am concerned that there is no change context studied, limiting the information provided regarding the explanatory power of the Portuguese version of Openness to Change for change research.

Authors’ response 

Thank you for your suggestion. However, we clarify in the manuscript that the objective of the manuscript was not to study the levels of organizational change among workers in the different phases of an organizational change intervention, but to study the psychometric properties of the OTOCS instrument.

Given the use of the original Miller et al. [7] scale (and other similar scales) in both change contexts and non-change contexts, the present study provides evidence of the OTOCS’ psychometric properties and responses from the two country samples.

We also suggest that future research must approach the predictive evidence of the OTOCS.

###########################################################

Reviewer 2’s comments

###########################################################

I also am concerned with idiosyncratic adjustments made in the interdependencies among items in order to engineer good fit. I recognize that tools are available in various structural equation programs to enhance conventional indicators, but this is a practice not widely accepted. It would be better to report the non-engineered and engineered indicators of fit and talk through the sensitivity of observed effects to these adjustments.

Authors’ response 

In an effort for the reader to understand the effort to discern the best fit of the data, the manuscript reports the initial analysis and subsequent modification indices both in terms of effect size and statistical significance:

“The original OTOCS structure provided a mediocre fit for data obtained with the merged samples (χ2(5) = 319.298, p < .001, χ2/df = 63.860; n = 1,175; CFI = .918; NFI =.917; TLI = .837; SRMR = .105; RMSEA = .231; P(rmsea) ≤ .05) < .001, 90% CI ].210; .253[). The modification indices were investigated, and a correlation path between item’s 2 and item’s 4 residuals was added to the model (r = .459, p < .001). The modified model presented an acceptable fit to the data from the merged samples (χ2(4) = 96.363, p < .001, χ2/df = 24.090; n = 1,175; CFI = .976; NFI =.975; TLI = .940; SRMR = .058; RMSEA = .140; P(rmsea) ≤ .05) < .001, 90% CI ].117; .165[).”

Readers should be able to judge the appropriateness of the added path, given the correlation between the two items. Nevertheless, we added emphasis to it in the Discussion.

“Adding a-theoretical paths based on modifications indices is inappropriate [99] however the included modification is justifiable by the fact that both items belong to the same factor (residuals might be associated) and due to both items being negatively worded [100]. The use of reversed items has its trade-offs, and its use should be cautioned [100,101].”

###########################################################

Reviewer 2’s comments

###########################################################

So, I am suggesting that a connection between organizational change and the present survey and sample design is not yet established. Can you make a better case, perhaps through additional change-related measures, or further evidence regarding the change experience of your sample?

Authors’ response 

We appreciate your perspective and effort to strengthen the study by positioning the relevance of the OTOCS in light of a current or impending organizational change.

We also suggest, as noted earlier, there is value in measuring employees’ openness to change when a planned change is not ongoing or imminent.

To quote Malik and Garg [5] regarding the importance of studying employees’ attitude toward change in IT companies, “technological transformations put immense pressure on employees since they encounter frequent changes in roles and responsibilities and reskill constantly to gear up with new lines of reporting and modifications in IT operations. This necessitates these firms to prepare their employees to adopt to change rather than retaliate against it. IT organizations thus need to ensure that their employees adapt effectively to these change processes and elicit affective commitment to change.”

###########################################################

Reviewer 2’s comments

###########################################################

I suggest as an alternative a focus on the Portugeuse language assessment of common organizational survey measures. I do not know this literature but in trying to find contributions that can be made by the current data set that possibility comes to mind.

Authors’ response 

Thank you for your suggestion to enhance the potential contribution of this data set to Portuguese-speaking individuals and organizations. We will seek to incorporate this idea in future studies.

+++++++++++++++++++++++++++++++++++++++++++++++++++++++++++

References

1. MacCallum RC, Browne MW, Sugawara HM. Power analysis and determination of sample size for covariance structure modeling. Psychological Methods. 1996;1: 130–149. doi:10.1037/1082-989X.1.2.130

2. Chen F, Curran PJ, Bollen KA, Kirby JB, Paxton P. An empirical evaluation of the use of fixed cutoff points in RMSEA test statistic in structural equation models. Sociological Methods & Research. 2008;36: 462–494. doi:10.1177/0049124108314720

3. Maydeu-Olivares A, Shi D, Rosseel Y. Assessing Fit in Structural Equation Models: A Monte-Carlo Evaluation of RMSEA Versus SRMR Confidence Intervals and Tests of Close Fit. Structural Equation Modeling: A Multidisciplinary Journal. 2018;25: 389–402. doi:10.1080/10705511.2017.1389611

4. Shi D, Maydeu-Olivares A, Rosseel Y. Assessing fit in ordinal factor analysis models: SRMR vs. RMSEA. Structural Equation Modeling: A Multidisciplinary Journal. 2020;27: 1–15. doi:10.1080/10705511.2019.1611434

5. Malik P, Garg P. The relationship between learning culture, inquiry and dialogue, knowledge sharing structure and affective commitment to change. Journal of Organizational Change Management. 2017;30: 610–631. doi:10.1108/JOCM-09-2016-0176

6. Chai DS, Song JH, You YM. Psychological ownership and openness to change: The mediating effects of work engagement, and knowledge creation. Performance Improvement Quarterly. 2020;33: 305–326. doi:10.1002/piq.21326

7. Miller VD, Johnson JR, Grau J. Antecedents to willingness to participate in a planned organizational change. Journal of Applied Communication Research. 1994;22: 59–80. doi:10.1080/00909889409365387

8. Danish RQ, Asghar J, Ahmad Z, Ali HF. Factors affecting “entrepreneurial culture”: The mediating role of creativity. Journal of Innovation and Entrepreneurship. 2019;8: 1–12. doi:10.1186/s13731-019-0108-9

9. Wanberg CR, Banas JT. Predictors and outcomes of openness to changes in a reorganizing workplace. Journal of Applied Psychology. 2000;85: 132–142. doi:10.1037/0021-9010.85.1.132

10. Herscovitch L, Meyer JP. Commitment to organizational change: Extension of a three-component model. Journal of Applied Psychology. 2002;87: 474–487. doi:10.1037//0021-9010.87.3.474

11. Gori A, Topino E. Predisposition to change is linked to job satisfaction: Assessing the mediation roles of workplace relation civility and insight. International Journal of Environmental Research and Public Health. 2020;17: 1–16. doi:10.3390/ijerph17062141

12. di Fabio A, Gori A. Developing a new instrument for assessing acceptance of change. Frontiers in Psychology. 2016;7: 1–10. doi:10.3389/fpsyg.2016.00802

13. American Educational Research Association, American Psychological Association, National Council on Measurement in Education. Standards for Educational and Psychological Testing. Washington, DC, USA: American Psychological Association; 2014.

---

## [Decision Letter · Decision Letter 1]

30 Mar 2021

Openness Toward Organizational Change Scale (OTOCS): Validity Evidence from Brazil and Portugal

PONE-D-20-29649R1

Dear Dr. Marôco,

We’re pleased to inform you that your manuscript has been judged scientifically suitable for publication and will be formally accepted for publication once it meets all outstanding technical requirements.

Kind regards,

Mariagrazia Benassi

Academic Editor

PLOS ONE

Additional Editor Comments (optional):

Reviewers' comments:

Reviewer's Responses to Questions

**Comments to the Author**

1. If the authors have adequately addressed your comments raised in a previous round of review and you feel that this manuscript is now acceptable for publication, you may indicate that here to bypass the “Comments to the Author” section, enter your conflict of interest statement in the “Confidential to Editor” section, and submit your "Accept" recommendation.

Reviewer #1: All comments have been addressed

Reviewer #2: All comments have been addressed

2. Is the manuscript technically sound, and do the data support the conclusions?

Reviewer #1: Yes

Reviewer #2: Yes

3. Has the statistical analysis been performed appropriately and rigorously? 

Reviewer #1: Yes

Reviewer #2: Yes

4. Have the authors made all data underlying the findings in their manuscript fully available?

Reviewer #1: No

Reviewer #2: Yes

5. Is the manuscript presented in an intelligible fashion and written in standard English?

Reviewer #1: Yes

Reviewer #2: Yes

6. Review Comments to the Author

Reviewer #1: Dear authors, I have appreciated your improvments and think the paper is fair to be published.

Best Regards

Reviewer #2: I accept your responses to my concerns and am persuaded by your arguments. I have no further comments.

7. PLOS authors have the option to publish the peer review history of their article (what does this mean?). If published, this will include your full peer review and any attached files.

Reviewer #1: No

Reviewer #2: No

---

## [Editor Report · Acceptance letter]

6 Apr 2021

PONE-D-20-29649R1 

Openness Toward Organizational Change Scale (OTOCS): Validity Evidence from Brazil and Portugal 

Dear Dr. Marôco:

I'm pleased to inform you that your manuscript has been deemed suitable for publication in PLOS ONE. Congratulations! Your manuscript is now with our production department. 

Kind regards, 

on behalf of

Dr. Mariagrazia Benassi 

Academic Editor

PLOS ONE